# The Use of Reproductive Indicators for Conservation Purposes: The Case Study of *Palinurus elephas* in Two Fully Protected Areas and Their Surrounding Zones (Central-Western Mediterranean)

**DOI:** 10.3390/biology11081188

**Published:** 2022-08-07

**Authors:** Cristina Porcu, Laura Carugati, Andrea Bellodi, Pierluigi Carbonara, Alessandro Cau, Danila Cuccu, Faustina Barbara Cannea, Martina Francesca Marongiu, Antonello Mulas, Alessandra Padiglia, Noemi Pascale, Paola Pesci, Maria Cristina Follesa

**Affiliations:** 1Dipartimento di Scienze della Vita e dell’Ambiente, Università degli Studi di Cagliari, Via Tommaso Fiorelli 1, 09126 Cagliari, Italy; 2COISPA Tecnologia & Ricerca, Stazione Sperimentale per lo Studio delle Risorse del Mare, Via dei Trulli 18, 70126 Bari, Italy; 3Dipartimento di Scienze della Vita e dell’Ambiente-Macrosezione Biomedica-Università di Cagliari, Cittadella Universitaria di Monserrato, 09042 Monserrato, Italy

**Keywords:** European spiny lobster, fecundity, reproductive output, egg production, no-take zones

## Abstract

**Simple Summary:**

The European spiny lobster *Palinurus elephas* is a species with a high commercial value that inhabits the Mediterranean Sea and the adjacent Atlantic waters and is classified by IUCN as vulnerable due to its continuous overfishing. In this study, we analyse the reproductive parameters of *P. elephas* populations in two different fully protected areas, including their surrounding commercial zones, in Sardinia (Italy, central-western Mediterranean), where a restocking programme was carried out. Here, data on fecundity, size at maturity, vitellogenin concentration and temporal trends of egg production of *P. elephas* was provided, and the relevance of this information for fisheries management and conservation planning is discussed.

**Abstract:**

In 1990s, the European spiny lobster *Palinurus elephas*, one of the most commercially important species in the Mediterranean, exhibited a population decline. For this reason, fully protected areas (FPAs) appeared effective in re-establishing natural populations and supporting fishery-management objectives. Here, the reproductive parameters of *P. elephas* populations in two different FPAs (Su Pallosu and Buggerru, central-western Mediterranean), where a restocking programme was carried out, and in their surrounding commercial zones, were investigated from quantitative and qualitative perspectives. The comparison of fecundity between females collected inside and outside FPAs did not show statistical differences as well as the vitellogenin concentration, which did not vary among eggs of different size classes of females caught inside and outside the FPAs, indicating the same reproductive potential. The study demonstrated a benefit of overexploited populations in terms of enhancement of egg production overtime (15 years for Su Pallosu and 6 years for Buggerru) with a mean egg production 4.25–5.5 times higher at the end of the study than that observed at the beginning of the study. The main driver of eggs production appeared to be size, with larger lobsters more present inside the FPAs than outside. Given these results, the dominant contribution of the two studied FPAs to the regional lobster reproduction is remarkable.

## 1. Introduction

Achieving the target of sustainable exploitation of the species subject to fishing is a crucial objective for fishery management [1,2]. However, despite the great amount of information and knowledge accumulated during the last decades, in some cases, traditional management tools have proven insufficient to prevent overfishing of target and bycatch populations [3]. Among the conservation tools aimed at re-establishing the good status of a resource, the establishment of fully protected areas (FPAs), equivalent to no-take marine reserves where any consumptive activity on marine stocks is prohibited differently from the partially protected marine protected areas that allow extractive activities to different degrees, is widely recognized as the most reliable [3]. Indeed, FPAs can be effective tools for both re-establishing natural populations and supporting fishery-management objectives. FPAs can help biomass to rebuild and to retain the spawning stock of heavily fished species [4,5] by suppressing the harvesting of reproductive animals and allowing retained animals to grow to a larger size [6,7]. Several studies have described a positive correlation between the increase in female age-size and larvae survival in fish (e.g., [8,9,10]), suggesting that the maternal effect (i.e., maternal size) on egg quality is very important for recruitment success [11]. In marine invertebrates, such as crustaceans, it was observed as the egg production rises with increasing size of females inside the FPAs, and fishing in adjacent areas may improve due to the spillover of both adult individuals and larvae [12,13,14,15], leading to a greater reproductive fitness, not only from a quantitative (fecundity and number of eggs) but also from a qualitative perspective (eggs’ dimensions and vitellogenin concentration) [16]. 

Among the processes involved in the reproduction, in recent years, vitellogenin (VTG) has been recognized as a biomarker in vertebrates, thanks to its ability to undergo variations of gene expression linked with the environmental condition [17,18,19], and/or a useful tool for monitoring the reproductive success of a species living in a specific environment [20,21]. Particular attention has been given to assessing the adverse effects of stressors, such as pollutants, that interfere with the endocrine system (endocrine-disrupting chemicals) in aquatic environments [22]. However, fishing pressure may also constrain the reproductive potential following two key routes: (i) fishing pressure modifies population demography by removing the bigger specimens; (ii) fishing removes individuals at various trophic levels of the ecosystem, affecting the distribution of energy and hence the amount of energy available for each individual, including the energy needed for reproduction [23,24]. 

Lobsters belonging to the family Palinuridae are among the most highly priced seafood in the world [15,25]. Among them, the European spiny lobster, *Palinurus elephas* (Fabricius, 1787), is the most commercially important species in the Mediterranean and North-Eastern Atlantic [26,27]. In the European market, it can be sold for prices ranging between EUR 40 and EUR 120 per kg [28]. However, as a general trend, in the second half of the 20th century, *P. elephas* populations faced a severe decline of its biomass at sea [15,29], due to the excessive fishing that had depleted its populations [29]. For this reason, the species is assessed as “vulnerable” by the International Union for Conservation of Nature [30]. At the beginning of the previous century, in the Mediterranean, *P. elephas* was captured by means of traps/pots and sometimes by scuba diving [31,32,33,34]. A transformation of the exploitation strategy took place during the 1960s and 1970s, with the progressive introduction of trammel nets. This change in fishing strategy had an impact on exploitation levels, demography, and sex composition of the populations which, being strongly influenced by the selectivity of the gear, led to overexploitation. In this worrying picture, a few regulations, leading to an increase in the stock, were implemented by the European Union and Italian Government [35]. However, the need for innovative management policies focusing, together with those already implemented, on reversing the decline of the stock, appeared essential. In this perspective, several no-take marine areas were established in central-western Sardinia, in 1998 firstly and in 2009 subsequently [15,35,36,37], as areas of restocking to protect lobster resource. Results of the implementation of more recent FPAs (in 2009) showed a burst in the variations in terms of catch per unit effort (CPUE) in density and biomass of European spiny lobster stocks (by ca. 300–700%) just 2 years following FPAs establishment, while also reporting tangible spillover effects (ca. 30–50% increase in density and biomass CPUE outside the FPAs) by the end of the program [15]. Instead, data from a 15-year continuous monitoring of a pilot FPA established in 1998, where the restocking protocol was conducted for the first six years and protection kept in force once restocking ceased, demonstrated the persistence in terms of restocking benefits [15,35,36,37].

Given these results and considering that differences in reproductive output of lobsters populations within and outside FPAs have been assessed in several areas (e.g., [5,14]), the aim of this paper was to analyse the reproductive potential of female spiny lobsters as a criterion for evaluating the effect of restocking and the cessation of fishing on egg production of *P. elephas*, in two Mediterranean FPAs (Sardinian western coast), after different periods of implementation, and their contribution as tools for lobster conservation. More specifically, we (1) investigated the reproductive fitness of the *P. elephas* populations in these two different FPAs and in their surrounding commercial zones from both a quantitative (fecundity and egg production) and a qualitative perspective (VTG levels in egg clutches and eggs’ dimensions) and (2) compared the reproductive potential of both FPAs and their surrounding zones in order to underline the efficiency of FPAs in the conservation of species. To the best of our knowledge, this is the first study to use the qualitative analysis of yolk-protein synthesis as an indicator of female reproductive activity in an invertebrate species (i.e., spiny lobsters) for conservation purposes.

## 2. Materials and Methods

### 2.1. Study Area and Sampling 

The study was conducted in the framework of a restocking programme of the European spiny lobster in Sardinian Seas, promoted and funded by the Autonomous Region of Sardinia (Regional Law No. 776 of 5 June 1998 and No. 82069/DecA/84, 8 November 2009). The first FPA (area 4.0 km^2^) was established in 1998 on the central-western coast (Su Pallosu) (see [38,39] for a description of the FPA), whereas the second (area 7.6 km^2^) was established on the south-western coast (Buggerru) in 2010 [15,37] (Figure 1). Both FPAs were established over rocky bottoms characterised by coralligenous and precoralligenous habitats [40,41].

Restocking was carried out effectively from 1998 to 2005 in the Su Pallosu FPA and from 2011 to 2015 in the Buggerru FPA through a Collaborative Fishery Research project [15], by which fishermen actively assist scientists in all phases of research. Specimens used for restocking were caught during commercial fishing in areas nearby and outside the FPAs. Restocked females were measured by the researchers, and if they fell below the minimum landing size (MLS, 90 mm of CL), they were tagged immediately and released in the centre of the reserve, at the end of the fishing day. Specimens above the MLS were kept by the fishermen to be sold in the fish market.

### 2.2. Sample Collection

The captures of *P. elephas* females within (IN) and outside (OUT) the FPAs were conducted annually from 1998 to 2013 in Su Pallosu at depths of 50–80 m and from 2010 to 2015 at Buggerru at depths of 40–70 m. The surveys were conducted, following a random sampling design, by the crew of commercial boats with the same gear used (trammel nets with a nominal mesh size ranging from 50 to 73 mm) in the commercial catches usually performed outside FPAs. The scheme of sampling, used in the two FPAs, was reported in [15,35].

Based on previous studies [15,39], data from outside FPAs were comprised within 5 nautical miles from the FPA centre. Such distance has been set as it exceeds the theoretical maximum distance that a spiny lobster could reach from the release point at the centre of the FPA. 

For each female, carapace length (CL, mm) was recorded to the nearest 1 mm below using a calliper. The presence or absence of external eggs were also recorded.

Table 1 presents a summary of the surveys and the number of released female spiny lobsters sampled in each year of restocking, from 1998 to 2005, at Su Pallosu, and from 2011 to 2015, at Buggerru. After 2005, in Su Pallosu FPA, the release of lobsters was stopped because the carrying capacity of the area was reached [35].

### 2.3. Egg Stage, Number and Dimension 

To analyse stage, number and dimension of eggs, berried females were caught in each FPA and in surrounding fishing areas during experimental surveys conducted ad hoc from October to March, corresponding to the egg extruding period of the species [42]. For each female, the eggs were divided into three development stages (stage1, stage 2 and stage 3) according to the scale proposed by [43], modified ad hoc for the species, based on the presence of visible eye spots and on the egg mass colour (Appendix A). Prior to preparation for any egg elaboration, excess liquid was filtered off, and the egg masses were placed on absorbent towels [42]. The egg total weight was determined to the nearest 0.0001 g with an electronic balance. A subsample of clutches from each maturity stage of berried females was measured, recording the minimum and maximum diameter, through the image analysis software tpsDig2 [44]. Differences between mean egg sizes at different stage were assessed by *t*-test [45], while the relationship between egg size in each maturity stage and lobster size was assessed using regression analysis, and statistical differences were evaluated using analysis of covariance (ANCOVA) with lobster size as a covariate [46].

The relative egg weight (REW) was obtained using the following formula [47]:REW = (EW/EN) × 10^5^(1)
where EW is the weight of each subsample and EN the number of eggs counted in each subsample. This index is directly related to egg weight: the greater the index, the heavier the individual eggs. REW was compared among the three developmental stages (*t*-test) and female size (ANCOVA). 

In addition, the number of eggs (at stages 1, 2 and 3) per body gram according to size was estimated and tested (ANCOVA).

### 2.4. Fecundity Estimation

Fecundity (total number of eggs carried externally on the pleopods) was performed first for clutches of the three maturity stages separately and then for all the clutches considered together.

Three random subsamples of the female’s total egg mass were weighed and separated for processing and counting. Then, two observers counted eggs manually using a 10× desk magnifier.

The parameters of the function relating fecundity to body size (IN and OUT separately for each FPA) were estimated using linear regression analysis. ANCOVA test was performed to compare fecundity within and outside the FPAs using lobster size as a covariate [45].

Subsequently, body sizes were converted to ages using the von Bertalanffy growth curve parameters calculated for *P. elephas* females (CL∞ = 120.2 ± 11.51 SE, K = 0.21 ± 0.05 SE, t0 = –0.349) by [48] using tagged, released and recaptured individuals, for which growth increments were analysed with two different procedures [49,50] contained in the Fisat package [51], thus obtaining the asymptotic carapace length (CL∞) and the curvature parameter (K) of the von Bertalanffy equation which were very similar to each other. The converted data (body sizes versus age) were used in estimating the age–fecundity relationship. Egg loss during incubation was also assessed by comparing fecundity–size equations from samples of berried females at the beginning (eggs at stage 1) and end (eggs at stage 3) of the incubation period [42]. The difference between the predicted fecundity values for a given size interval was used to calculate egg loss in that size range.

### 2.5. Functional Maturity

The presence of external eggs was used as an indicator of maturity. The size at functional maturity (L_50_), defined as the size at which 50% of females have eggs on pleopods, and the maturity range (MR = L_75_ − L_25_) were estimated, for each zone (IN and OUT) of the two FPAs and then for their totality, by fitting maturity ogive to the proportion of mature females in each 2 mm CL size class and using a binomial generalised linear model (GLM, [52]) with a logistic link [53]. Non-linear least-squares regression was used to estimate the parameters: p = 100 {1 + exp[a + b × CL]}^−1^(2)
where p is the proportion of mature females, a is the intercept, and b is the slope of the maturity curve. The length at maturity is L_50_ = (−a/b). 

In a similar way, a logistic model was fitted to the paired maturity and age data, and the age at which 50% of individuals have eggs on pleopods (A_50_), using the von Bertalanffy growth curve parameters calculated for *P. elephas* females by [48], was calculated using the following equation:p = 100 {1 + exp[a + b × Age]}^−1^(3)

Chen test [45] was performed to compare statically the ogives of maturity among unfished and fished areas and FPAs.

### 2.6. Relative Reproductive Potential

Data from 1122 females in Su Pallosu FPA and 359 in Buggerru FPA grouped in 5 mm CL classes, representing the population structure across the study period, were used to estimate the relative reproductive potential (RRP) to establish the size classes of breeding females contributing most to the egg production of the population within the FPAs. The RRP was estimated as follows:RRP = Ci × Mi × Bi(4)
where Ci is the proportion of the size class in all females, Mi is the proportion of ovigerous females in size class i, and Bi is the mean fecundity of females in size class i [42,54].

### 2.7. Vitellogenin Concentration

In 2013, VTG concentration was estimated from egg clutches of 16 ovigerous females caught within both FPAs (60.9–91.9 mm CL) and 14 caught outside both FPAs (68.7–100.3 mm CL). Specimens with eggs in the middle stage of development (stage 2, Appendix A) were selected considering that the VTG level in crustaceans seems to decrease significantly as the animals are near oviposition [55]. 

The VTG amount was determined via indirect ELISA (Enzyme-Linked Immunosorbent Assay) conducted using anti-VTG-specific primary antibodies for *P. elephas* [56]. The antibodies were created by Twin Helix (Twin Helix, Rho, Italy) starting from an immunogenic peptide synthesised by Genscript (Genscript USA Inc., Piscataway, NJ, USA). The primary structure of the immunogenic peptide was obtained in our laboratory from the partial cDNA sequence deposited in NCBI (GenBank: KX792013.1). For indirect ELISA, the commercial Prepro Tech TMB ELISA Buffer Kit (DBA, Italy) was employed following the manufacturer’s suggested protocol. For the calibration curve, ELISA plates were coated with the antigenic peptide diluted in 2% blocking medium in phosphate-buffered saline buffer (PBS, pH 7.4) at scalar concentrations in the range of 250-1 ng per well. Diluted anti-VTG-specific primary antibodies diluted at a concentration of 0.1 ug/well were incubated in the plates overnight, at 4 °C. An antimouse IgG conjugated with HRP (DBA, Italy) diluted in PBS (1:20,000) was used as a secondary antibody. Absorbance values (A460 nm) were measured after 2 h of enzyme–substrate incubation with tetramethylbenzidine at room temperature. The calibration curve coefficient value obtained (r^2^ = 0.9757) revealed a good correlation between the scalar concentrations of antigenic peptide and OD at 460 nm. The quantification of VTG was determined using *P. elephas* eggs. For this purpose, ELISA plates were coated with 25 mg of eggs homogenate per well, obtained by sonication in RIPA buffer (BIOLAB, Cagliari, Italy). The samples were processed following the same steps previously described for the antigenic peptides used for the standard curve.

### 2.8. Index of Egg Production

The annual CPUE for mature female spiny lobsters (assuming a knife-edge maturity with size at functional maturity) captured within and outside each of the two FPAs before their establishment in 1998 (Su Pallosu) and 2010 (Buggerru), and for the subsequent 15 (Su Pallosu) and 5 (Buggerru) annual fishing seasons (March 1–September 30), was calculated. The CPUE was expressed as the total number of mature female lobsters caught per piece of net (i.e., 50 linear meters of net). Temporal trends of CPUE were analysed through non-parametric permutational analyses of variance ‘PERMANOVA’ (software PRIMER 7, Plymouth Marine Laboratory), based on ‘Euclidean distance’ similarity matrixes of non-transformed data [57]. The experimental design included 2 orthogonal fixed factors: time (years) and FPA (with 2 levels: inside vs. outside), with a variable number of replicates. Post hoc tests were also conducted to assess: (i) differences in CPUE between inside and outside the FPA across the 15 years for Su Pallosu FPA and five years and (ii) differences in CPUE across sampling years, within inside and outside the FPA, separately.

Temporal size–frequency distributions of female *P. elephas* inside and outside Su Pallosu (data combined in two years) and Buggerru (data reported for each year) FPAs was reported standardized to CPUE (number of females at each size class, 5 mm CL, caught per piece of net)) and the temporal trends were tested using the non-parametric permutational analyses of variance ‘PERMANOVA’ (see above). 

A standardized index of egg production (IEP) was estimated using CPUE of mature females, separately for each FPA (IN and OUT), with the summation of the contribution of each CL class of 5 mm:Σ (CPUECL × fecundityCL)(5)
where CPUECL is the mean catch of mature females at each size class (5 mm CL) caught per piece of net and fecundityCL is the fecundity at each mature size class (5 mm CL), using the fecundity–CL relationship here estimated. All fecundity values used were corrected for egg loss.

Annual trends in IEP between 1998 and 2013, at Su Pallosu, and between 2010 and 2015, at Buggerru, both within and outside the FPAs, were estimated and compared using a *t*-test repeated for each area and period.

## 3. Results

### 3.1. Egg Stage, Number and Dimension

Analysing the relationship between the mean diameter of the eggs by maturity stage and female size, eggs diameter does not appear related to the female dimension (63.6–100.4 mm) (ANCOVA, *p* = 0.83), but rather to their developmental stage of the berried females: higher the stage bigger the difference. Therefore, stage 1 (*n* = 1977, mean ± SD = 1.07 ± 0.06 mm, range = 0.94–1.13) and stage 2 egg diameters (*n* = 832, mean ± SD = 1.12 ± 0.07 mm, range = 0.92–1.18) were significantly smaller than stage 3 egg diameters (*n* = 1342, mean ± SD = 1.25 ± 0.09 mm, range = 1.05–1.35) (*t*-test, *p* < 0.001). The same was registered for the REW. It increased from stage 1 to stage 3 (stage 1 = 77.01 ± 10.79; stage 2 = 82.22 ± 5.63; stage 3 = 98.53 ± 15.27), revealing significant differences in dimension among the embryonic stages (*t*-test, *p* = 0.003). In line with the observations regarding the diameter, no significant difference in REW with female size (ANCOVA, *p* = 0.52) was found.

The number of eggs per body gram ranged from 34.5 to 150 eggs (Figure 2) and a statistically significant relationship with body size was found (ANCOVA, *p* = 0.009), indicating that maximum relative fecundity occurred in the largest berried females as follows: number of eggs for body gram = 0.0165*CL^2^ − 1.9685*CL + 144.01, R^2^ = 0.196).

### 3.2. Fecundity Estimation 

Fecundity data were available for a total of 130 female spiny lobsters ranging between 63.6 and 100.4 mm of CL (3–9 y). 

The fecundity–size relationships (Table 2; Figure 3a,b) did not differ between specimens collected inside and outside each FPA (Su Pallosu, ANCOVA, *p* = 0.034; Buggerru ANCOVA, *p* = 0.026), and thus, data were pooled together. 

Considering the whole sample (Su Pallosu and Buggerru together), the number of eggs in a single brood varied from 13,964 in the smallest lobster to 112,417 in the biggest one and the relationship between CL (and age) and number of ovo-deposited eggs (F), was estimated as F = 2245 × CL − 135,898 (r^2^ = 0.80) and F = 16,560 × age − 38,515 (r^2^ = 0.83) (Figure 3c,d). 

Egg loss was determined by comparing fecundity–size relationships within that size range at the beginning (eggs at stage 1) (F = 2577.4 × CL − 156,990, r^2^ = 0.71) and end (F = 2565 × CL − 156,093, r^2^ = 0.76) of the incubation period (eggs at stage 3) with mean fecundity at the beginning being slightly greater than at the end of the incubation period. The distance between the two fitted lines at the mid-point (between 80 and 85 mm CL) indicated a mean egg loss of 14% of the fecundity of newly berried females of that size.

### 3.3. Functional Maturity 

Inside the Buggerru and Su Pallosu FPAs, the estimated L_50_ of *P. elephas* ovigerous females was 81.2 mm and 85.4 mm CL, respectively (Table 3), corresponding to an age at maturity (A_50_) of 5.0 and 5.5 years, respectively. Outside FPAs, the L_50_ were very similar to those observed inside (Buggerru OUT L_50_ 80.0 mm CL (A_50_ = 4.9 years) and Su Pallosu OUT L_50_ 83.0 mm CL (A_50_ = 5.24 years) (Table 3), not showing significant differences among them (Chen-test, *p* = 0.42).

L_50_, obtained pooling data inside and outside for each FPA, was 81.8 mm CL for Buggerru and 84.8 mm CL for Su Pallosu (not statistically different, Chen test, *p* = 0.33), while L_50_ considering the two areas (Buggerru + Su Pallosu) together was 82.6 mm CL (Table 3, Figure 4).

### 3.4. Relative Reproductive Potential 

Within the Su Pallosu FPA, the smallest ovigerous female was 63.6 mm in CL. The CL classes between 65 and 75 mm had the greatest percentage of females (Figure 5a). In the Buggerru FPA, the smallest ovigerous female was 64.5 mm CL and the CL classes between 75 and 80 mm had the greatest percentage of females (Figure 5b).

The size classes that contributed to maximum RRP in both FPAs were fully below the minimum landing size (MLS) established for the species (90 mm CL). Inside the Su Pallosu FPA, the maximum RRP was attributed to the 75 mm CL size class (below it was 20% of the RRP), and size classes below the MLS contributed 68% of the RRP (Figure 5a). The smallest (60–70 mm CL) and largest (>100 mm CL) ovigerous females showed low values of the RRP. Females below the L_50,_ calculated for the FPA analysed (85.4 mm CL) and representing 87% of the female population considered, contributed 58% of the RRP. 

Regarding the Buggerru FPA, the maximum RRP was registered for the 85 mm CL size class (below it was 23.5% of the RRP), and 88.5% of egg production was registered by size classes below the MLS (Figure 5b). Females below the L_50_ (81.2 mm CL), relative to the Buggerru FPA (representing 88% of the female population considered), contributed 58% of the RRP.

### 3.5. Vitellogenin Concentration

By relating the absorbance of each of our samples to unknown concentration with the standard curve, we obtained the concentrations (ng/mL) of VTG in the eggs of *P. elephas*, all of development stage 2. These were found to be quite high (in a range of 200–256 ng) in 12 of the 30 individuals analysed (Figure 6a). In 16 individuals, the concentrations were within the range 120–170 ng/mL, whereas only two individuals exhibited very low values compared with the previous groups (Figure 6a).

Moreover, the analysis of the VTG concentration in eggs at stage 2 (Table 4) in relation to CL within and outside the FPAs did not show statistically significant differences among slopes (ANOVA, F-ratio = 1.04, df = 1, *p* = 0.39) (Figure 6a). 

Considering the whole sample, VTG concentration did not vary among the size classes analysed (5 mm CL) (ANOVA, F-ratio = 0.42, *p* = 0.8715) (Figure 6b).

### 3.6. Index of Egg Production (IEP)

Catch per unit effort (CPUE) of mature females varied between inside and outside each FPA and with time (Figure 7; Appendix A). 

In the Su Pallosu FPA, the IEP episodically oscillated from 1998 (time 0) to 2004 as a consequence of the female CPUE variations (Figure 8a). After six years, even if the restoking stopped as the carrying capacity in the FPA was reached, a significant increase in the index was observed, due to an increase in size of females which, achieving the size at maturity, registered a higher reproductive activity with a consequent increase in abundance within the FPA (Appendix A) (Figure 8a and Appendix A). At the end of the study, the IEP was 4.25 times higher than that at the beginning. In the final period (2011–2013), a slight decrease was registered even though the IEP always stayed higher than the mean value registered in the IN zone. In the Su Pallosu surrounding area (OUT), from 2007, the IEP registered an increasing trend, even though it was lower than that within the FPA (Figure 8a). Rates of the annual egg production enhancement varied, ranging between 68% and 474% and between 6% and 230% within and outside the FPA, respectively.

Within the Buggerru FPA, the IEP increased and was higher overtime (Appendix A) than that registered in the neighbouring areas. An abrupt increase was registered just two years (since 2012) after the FPA establishment with a slight decrease in the subsequent years (trend not significant, *t*-test, *p* > 0.05) (Figure 7b, Figure 8b and Appendix A). Within the FPA, it was 5.5 times higher in 2015 than in 2010. 

## 4. Discussion

An expected benefit of protecting overexploited populations is the enhancement of reproduction and subsequent increases in egg and larval export [58,59]. Currently, evidence of increased reproductive potential inside protected areas has been demonstrated [14,60], but few studies have shown this pattern in the adjacent fished areas [14,61].

Although lobster larval spillover is still not quantified for the Mediterranean FPAs, its role in sustaining local fisheries has been confirmed for *P. elephas*, from gradients of abundance in experimental and commercial catches along with tagging experiments in the Columbretes Islands [12] and Sardinia [15,35,36]. Moreover, few studies [42,61] have addressed the benefits of FPAs for populations and the fishery itself in terms of number of eggs produced by lobsters. To the best of our knowledge, our study is the first to address the efficacy of restocking the European spiny lobster in two FPAs, analysing the temporal trend of the egg production of *P. elephas* both within the FPA and in the neighbouring commercial area, from a quali-quantitative point of view. 

### 4.1. Fecundity

The fecundity–size relationship of *P. elephas* in Sardinian FPAs increases linearly, with the biggest females having highest fecundity values, as Atlantic [62] and Mediterranean [42,61,63] studies also reported in exploited populations, and not exponentially, as it has been demonstrated for the spiny lobsters of the genera *Panulirus* and *Jasus* [5,47,64,65,66,67]. However, a high degree of individual variability in fecundity has been recorded, a common pattern already observed in lobsters such as *P. elephas* in the western Mediterranean [42] and other spiny lobster species (e.g., [68]). Moreover, differently from the Mediterranean study [42], egg diameter does not appear to be related to females’ size but indeed to the stage of their development. 

Assessing egg loss provides an indication of the survival from fertilised eggs through hatching [42]. Egg loss appears to be directly proportional to the duration of the incubation period [46]. In this study, we estimated egg loss at 14% for females in the 80–85 mm CL size range, considered in the estimation of reproductive parameters. The degree of stress induced by the means of capture, such trammel nets, traditionally used in *P. elephas* Sardinian fishery, could be one of the main factors inducing the egg loss [26]. However, considering the egg loss rates described for other decapods and accounting for the length of the incubation period, our percentage is within the range reported for the other species (10–28%) [26,42,69]. 

Especially for long periods of protection (over a decade), the increase in fecundity in protected populations is maintained primarily by the large sizes of mature females rather than the overall abundance of the mature spiny lobster [70]. Indeed, no difference has been found in the fecundity–size relationship of lobsters caught within and those caught outside the FPAs. This result has been confirmed from a biochemical point of view as well. The eggs (of the same stage of development) of specimens caught within the FPAs and those of specimens caught in their neighbouring commercial areas do not exhibit statistically significant differences in VTG quantity. Moreover, VTG concentrations did not varied among size classes of mature females caught in the different zones (IN and OUT), indicating that each egg would have the same potentiality in terms of reproductive fitness (survival of the larvae and life expectancy of juveniles. The same VTG levels found in eggs of *P. elephas* individuals grown in both restocking and surrounding fishing areas could represent useful information showing no changes in protein levels, due to the presence of oestrogen-like substances in the water in different areas [56].

### 4.2. Size at Maturity

Sizes at functional maturity (L_50_), estimated inside and outside FPAs, are very similar each other and not statistically different. The value of L_50_ estimated for Sardinian populations (82.6 mm CL) (considering Su Pallosu and Buggerru together) is similar to that reported by [71] for female *P. elephas* off Corsica (86 mm CL), and higher than that estimated by [42] in the Columbretes Islands Marine Reserve (77.2 mm CL). Several factors may be put forward to explain such differences in size at functional maturity of *P. elephas*. In spiny lobsters, size at maturity appears to be age specific (linked to growth rates) [43], and where growth is fast, sexual maturity would be reached at a greater size. Low population density could translate into lower competition for food [43] or shelter [72], thus allowing faster growth rates, which would rise to a larger size at maturity [73]. Moreover, larger size at maturity could also be environmentally driven [74,75,76]. However, it is well known that estimates of size at maturity differ depending on the maturity criteria used as well as the sampling period, number and size range of the specimens [75]. In our context, differences in L_50_ observed between Sardinian and Spanish females [42] could be ascribable to the different sampling period (Sardinia, from October to March; the Columbretes Islands, September).

### 4.3. Index of Egg Production 

Our results demonstrate a benefit of protecting overexploited populations in terms of enhancement of reproduction and subsequent increase in egg and larval export indicating that the egg production (meant as index of egg production) of *P. elephas* increases over time within both of the Sardinian FPAs. Indeed, at the end of the investigated period, values of 4.25 and 5.5 times higher than that registered at the beginning for the Su Pallosu and Buggerru FPAs, respectively, are registered. Rates of annual egg production enhancement range between 68% and 474% and between 6% and 230% within and outside the Su Pallosu FPA, respectively. 

It is known that an increase in female lobsters’ CPUE contributes to an increase in egg production over time [61,77,78]. For example, in the Columbretes Islands Marine Protected Area (Western Mediterranean), after 19 years of protection, mature females were on average 20 times more abundant, and egg production per unit area was 30 times greater in the MPA than in nearby fished areas [61]. In addition, in Northwestern Hawaiian Islands (Pacific Ocean), an increase of 16% in size-specific fecundity of the spiny lobster *Panulirus marginatus* after 14 years of exploitation was recorded, while [78] showed that five years after the coming into force of a Marine Protected Area in New Zealand in 1978, the population of the spiny lobster *Jasus edwardsii* had experienced a 4.5-fold increase in abundance. In our case study, the continuous recruitment of tagged small females (below the MLS of 90 mm CL) during the restocking [15,35], certainly influenced the increase in egg production, as well as growth rates and the maximum size of individuals. When the project started (1998 for Su Pallosu FPA and 2011 for Buggerru FPA), the demographic structure of the female population was unbalanced in favour of juveniles in both FPAs, with only few individuals above or near the size at maturity (L_50_). The rise in density of juvenile females due to the restocking should have influenced the progressive increase in fecundity over time, reaching a maximum RRP of 75 mm in CL within the Su Pallosu FPA and 85 mm in CL within the Buggerru FPA, sizes smaller than those registered in the Columbretes Island (western Mediterranean) [42,61]. 

Moreover, inside the Buggerru FPA, the positive effect on egg production is observed just two years after the establishment of the restocking program, in contrast to the Su Pallosu FPA, where this result is achieved after six years. It is important to stress how in Su Pallosu FPA, the effect provided by active restocking was discriminated by the single effect of protection, once restocking ceased, demonstrating the persistence in time of restocking’ benefits. These results are in agreement with [15] for the same FPAs, where CPUE, in terms of both density and biomass, burst (by ca. 300–700%) just two years after the establishment of the Buggerru FPA, while tangible spillover effects have been reported by the end of the programme [15]. 

The temporal shift in the reaching of abundance and egg-production increase in both FPAs suggests that factors other than time of protection and extension of the protected areas can contribute to the rise in lobsters’ egg production. The dynamic of commercial fishery before the establishment of the FPAs could be the primary driver of the temporal variations in lobster egg production observed in this study. Su Pallosu FPA, although smaller than Buggerru FPA (4 vs. 7.6 km^2^), took more time to restore its habitats and then to register an increased density, probably as a result of a stronger fishing effort exercised by the commercial fleet in the zone before its closure. The strong winds that blow on the southwest coast of Sardinia (where the Buggerru FPA is located) for most of the year force fishermen to refrain from fishing for long periods, thus safeguarding, at the same time, the environment [79]. This is indicated also by the RRP value, which, in the Buggerru FPA, has registered a higher value than in the Su Pallosu FPA in 5 years of protection (85 mm vs. 75 mm of CL). 

The index of egg production inside the Su Pallosu FPA increased over the 16 years of study, showing high values since 2006 (after 8 years of closure) and reaching the maximum IEP in 2010 (after 13 years). The stability of the mean egg production over the last three years may indicate that the size of lobsters, and therefore, the egg production may be approaching an asymptote, as has been observed in the Columbretes [61]. 

The main driver of egg production seems to be the size of the specimens—the bigger the females, the higher the egg production. Therefore, within the protected area, the progressively higher concentration of the largest mature females, more sedentary than the small and middle-sized ones [39], resulting from the protection, could be the main factor responsible for the increase in egg production. Even if the increase in density within the FPA (as a consequence of restocking and closure to fishing), could, on one hand, reduce the growth rates as a result of stronger competition for food, this would lead, on the other hand, to a greater concentration of large-sized specimens, with more fidelity to the site and more capacity in mating. In this sense, FPAs, where population demographic characteristics, such as size composition and sex ratio are expected to be restored towards the baseline conditions, are considered particularly valuable as reference systems when exploring fisheries effects on mating systems [80].

However, in both FPAs, females with a size larger than the L_50_ represent only about 13% of the population but contribute 42% of the egg production. Despite the poorer representation of the large females in our catches, they give an important contribution to the reproductive potential of the European spiny lobster population. In this perspective, the establishment of the FPAs as a refuge for the highest size/fecundity females, as shown in our study, could more effectively support stock recruitment. 

Finally, egg and larval export are very important in larval dispersal patterns and usually have greater potential than spillover of specimens to benefit exploited lobster populations [61]. Although the recruitment of locally spawned larvae to the Marine Protected Area (MPA) appears unlikely due to dispersal potential of *P. elephas* with a pelagic phase of 4 to 5 months [33,61], a growing body of evidence regarding fish larval dispersal patterns indicates that larval retention in, or larval return to, parental grounds is far more important than previously believed [7]. A recent study of the dispersal of *P. elephas* larvae in the southwest of England [81] highlighted that spiny lobster populations are capable of partial self-seeding. In any case, genetic studies have indicated that *P. elephas* in the Western Mediterranean are interconnected [82] and that regional populations may be linked via a common larval pool among intermediate populations. Based on these results, we underline the importance of managing spiny lobster fisheries across all their potential geographic range [83] and the need to manage species, such as *P. elephas*, in a way that considers their entire life cycle, prioritizing the stages of the cycle that could have the greatest benefits in terms of future recruitment. Improving our understanding of the life cycles and ecology of key fisheries species will strengthen our ability to successfully manage commercially important species. Accordingly, a rise in the number of MPAs within a network can increase the reproductive potential of the stock, with excellent results in terms of the sustainability of the resource. From this perspective, our experience proved how the dominant contribution of the two Mediterranean FPAs here analysed in terms of egg production in supporting to exploited populations was remarkable.

Certainly, future research is needed to increase the knowledge on potential larval export, pelagic larval duration in combination with local oceanographic circulation models that might provide new insights. At the individual level, the use of parentage assignment might be another option in a system where individuals are captured and recaptured (and eggs are also readily sampled—which might be used to verify parental assignment). The capture of individuals outside of the MPAs/FPAs could potentially be “back-tracked” to FPAs in a particle model, or assigned as offspring from known parents in an FPA in a genetic library, as demonstrated in *Hommarus gammarus* by [84]. In addition, male populations should be also considered because they are targeted disproportionately as a consequence of sexual dimorphism in spiny lobster size (i.e., males grow larger than females) [80,85] and because of protections for ovigerous females. For this reason, their overexploitation could lead to sperm limitation with serious repercussions for reproductive success also in *P. elephas*, as reported in several decapod populations (e.g., [80,85,86]).

Despite the existing low knowledge on larval dispersal, the results provided by this experience strengthen the importance of including FPAs in the management plans of resource. In our case, FPAs and their restocking, together with additive technical measures such as limiting the number of pieces of net or boats should be considered to further enhance the recovery of depleted stocks and their more sustainable exploitation on the long term.

## 5. Conclusions

In conclusion, our study demonstrates the importance of FPAs for maintaining optimal reproductive potential of spiny lobster populations by harbouring high abundances and mature size distributions. The contribution of the Mediterranean FPAs in terms of egg production in supporting exploited populations in commercial zone is consistent, even if further studies are needed to understand better the larval connectivity and dispersal model. 

## Figures and Tables

**Figure 1 biology-11-01188-f001:**
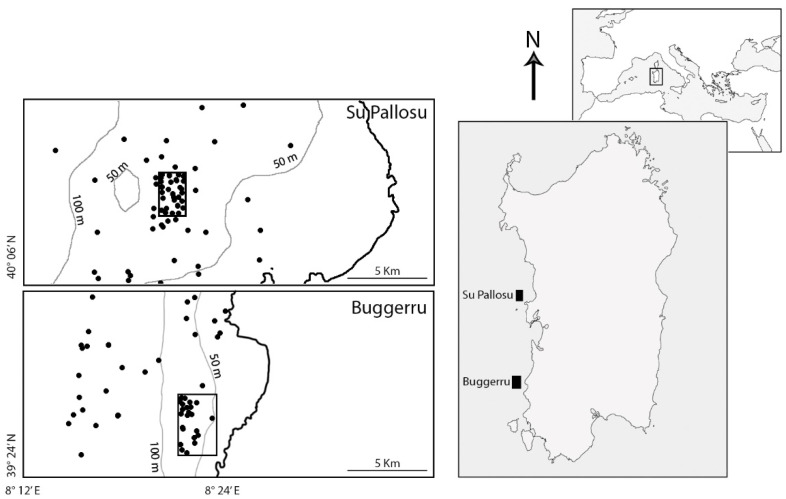
Map of the study areas. Dots inside the rectangles represent replicates (i.e., set of trammel nets) performed inside FPAs, where fishing activities are prohibited. Dots outside the rectangles represent set performed outside FPAs, in neighbouring fishing grounds.

**Figure 2 biology-11-01188-f002:**
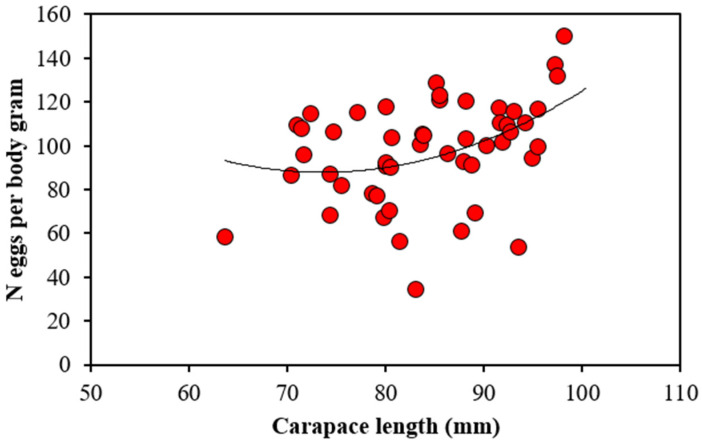
Number of eggs per body gram against Carapace length and fitted line for females *Palinurus elephas*. Equation of fitted line is given in the text.

**Figure 3 biology-11-01188-f003:**
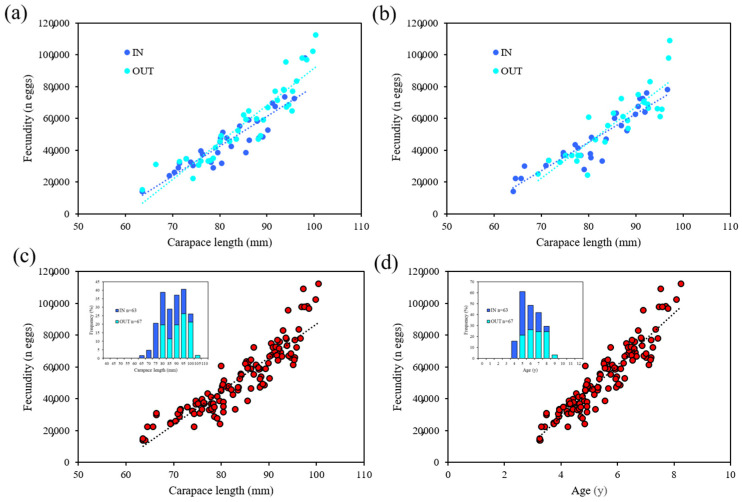
Fecundity–size relationships for *Palinurus elephas* based on data collected inside (IN) and in the neighbouring commercially fished zones (OUT) of (**a**) the Su Pallosu and (**b**) (Buggerru). Fecundity–size (**c**) and fecundity–age (**d**) relationships based on data collected inside of the two FPAs (Su Pallosu and Buggerru) and in the neighbouring commercially fished zones.

**Figure 4 biology-11-01188-f004:**
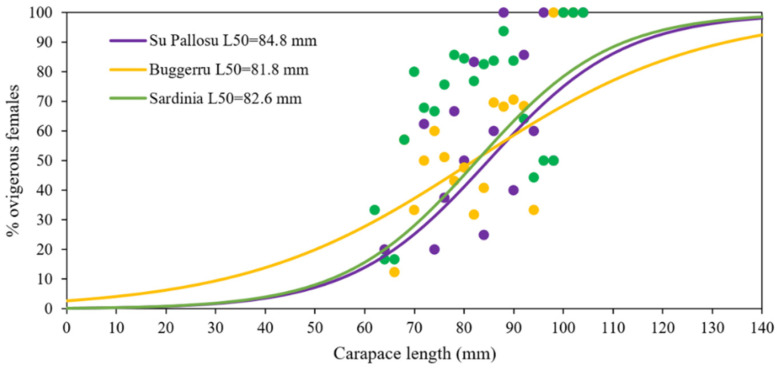
Proportion of *P. elephas* ovigerous females in each class of Carapace length (2 mm CL) estimated for Su Pallosu, Buggerru and the two areas combined together (Sardinia).

**Figure 5 biology-11-01188-f005:**
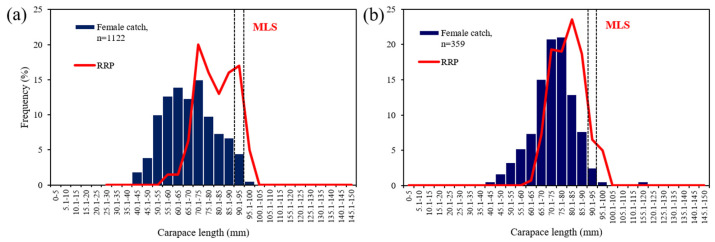
Relative reproductive potential (RRP) in relationship to carapace length plotted with the length–frequency distribution of the *Palinurus elephas* female catches in Su Pallosu (**a**) and Buggerru (**b**) FPAs. The dotted bar indicates the minimum legal size (MLS) of 90 mm CL.

**Figure 6 biology-11-01188-f006:**
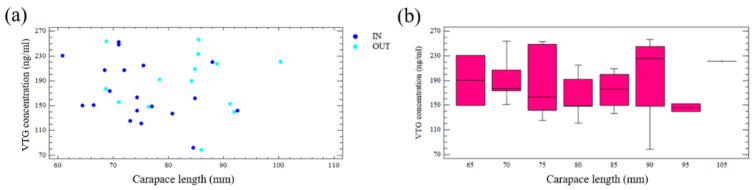
(**a**) Relationship between carapace length (mm) and relative VTG concentration of egg clutches at stage 2 of *P. elephas* females caught inside and outside FPAs (Su Pallosu and Buggerru); (**b**) mean VTG concentration of egg clutches at stage 2 of females in 5 mm CL size classes caught both inside and outside of the two FPAs together.

**Figure 7 biology-11-01188-f007:**
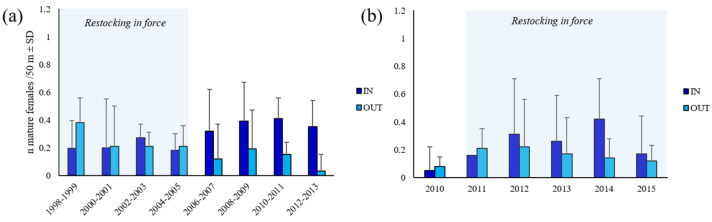
Plot showing (**a**) biannual mature female CPUE (*n*. females/50 m ± s.e.) trends inside and outside Su Pallosu and (**b**) annual mature female CPUE (*n*. females/50 m ± s.e.) trends inside and outside Buggerru. Pale blue rectangles indicate the restocking period.

**Figure 8 biology-11-01188-f008:**
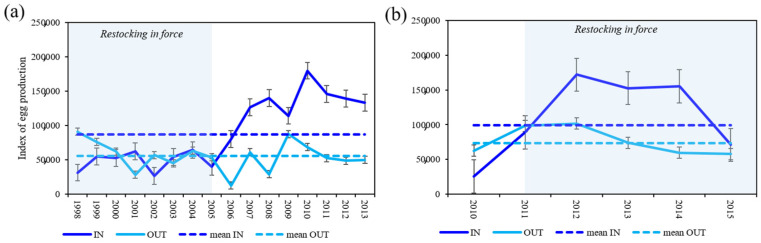
Annual trend of index of egg production (IEP) inside and outside Su Pallosu (**a**) and Buggerru (**b**). Means ± SD. Solid lines, IEP; stripped lines, mean IEP.

**Table 1 biology-11-01188-t001:** Depth range, number of surveys conducted inside and outside of Buggerru FPA by year (2010–2015) and Su Pallosu FPA (1998–2013) and number of released female spiny lobsters/inside of each FPAs.

FPA	Depth Range (m)	Year	N Fishing Sets	N F Released
Buggerru	40–70	2010	6	-
		2011	9	567
		2012	41	603
		2013	14	167
		2014	16	251
		2015	33	249
Su Pallosu	50–80	1998	22	479
		1999	40	324
		2000	39	450
		2001	33	173
		2002	32	35
		2003	26	9
		2004	32	199
		2005	26	63
		2006	24	-
		2007	36	-
		2008	186	-
		2009	57	-
		2010	47	-
		2011	110	-
		2012	101	-
		2013	32	-

**Table 2 biology-11-01188-t002:** Number and size ranges of berried females analysed for fecundity estimation calculated for the two Su Pallosu and Buggerru FPAs, separately, and their surrounding fishing areas.

FPA	N	Range CL (mm)	CL–Fecundity Relationship	r^2^
**Su Pallosu**				
IN	34	63.6–98.1	F = 1882 × CL − 108,073	0.84
OUT	38	66.4–100.4	F = 2414.8 × CL − 148,291	0.82
**Buggerru**				
IN	29	64.0–96.7	F = 1790 × CL − 98,190	0.85
OUT	29	71.5–97.2	F = 2227.6 × CL − 133,389	0.78

**Table 3 biology-11-01188-t003:** Size at functional maturity (L_50_), Maturity Range (MR), standard error (s.e.) are reported for Su Pallosu and Buggerru FPAs, inside (IN) and outside (OUT) and for the two areas combined together (Sardinia).

FPA	L_50_ (CL, mm) ± s.e.	MR ± s.e
**Su Pallosu**		
IN	85.4 ± 3.2	29.4 ± 8.6
OUT	83.0 ± 5.3	16.7 ± 6.4
IN + OUT	84.8 ± 4.1	20.3 ± 7.5
**Buggerru**		
IN	81.2 ± 19.9	25.6 ± 5.6
OUT	80.0 ± 14.0	23.4 ± 4.1
IN + OUT	81.8 ± 15.0	32.88 ± 6.1
**Sardinia**	**82.6 ± 1.63**	**26.4 ± 4.5**

**Table 4 biology-11-01188-t004:** Number of ovigerous females of *P. elephas*, their size range, eggs stage, vitellogenin (VTG) concentration range and mean estimated inside and outside FPAs.

Area	N	CL Range (mm)	Eggs Stage	VTG Range (ng/mL)	VTG Mean ± SD (ng/mL)
IN	16	60.9–84.8	2	120.93–252.71	177.01 ± 43.28
OUT	14	68.8–100.3	2	78.88–256.45	187.44 ± 49.37

## Data Availability

Data are contained within the article or Appendix A.

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
