# Peer review of "The Use of Reproductive Indicators for Conservation Purposes: The Case Study of Palinurus elephas in Two Fully Protected Areas and Their Surrounding Zones (Central-Western Mediterranean)"

_biology, 2022, doi:10.3390/biology11081188_

Round 1

Reviewer 1 Report

The egg sizes vary upon temperature and subsequent spawning in same intermoult period .

Sex ratio in FPA versus non FPA or commercial area is not discussed 

Egg loss could also be due to fertility and competition issues 

Reviewer 2 Report

This study attempts to evaluate the impact of protected areas on lobster reproductive potential. The authors completed a lot of work on measuring and analyzing multiple reproductive potential indicators. However, I do not think this study evaluates the impact of protected areas for a couple of reasons:

1.      The impact of restocking and the impact of the protected area cannot be separated. How can the study tell if the increase in reproductive potential is due to the FPA or restocking? I’m not sure how the impact of restocking can be removed. Because the impact of restocking is present in this study range, I do not believe this study is evaluating the impact of FPAs. It could evaluate the impact of FPAs and restocking together, but the paper seems to mainly focus on FPAs. Also, restocking occurred at different times in the two different FPAs so this effect would be difficult to disentangle.

2.      October to March is a long range, and lobster population dynamics could have shifted throughout this range. Catchability might have also changed.

3.      Catchability most likely differs between the different vessels, and this should be considered in the CPUE analysis. Month should also be considered in this analysis given the wide range of sampling (October to March).

Other comments:

-Some of the wording is awkward, and the manuscript should be carefully read through so that sentences can be edited.

-The paper should be checked for proper grammar.

Introduction:

-The introduction makes it seem like FPAs are not recognized as effective but then it is stated that they are effective. Which has the past literature showed?

-The introduction should highlight why this study is important. With the current text, it is not clear as to why this study is important.

-Why did the FPAs stop in 2013 and 2015?

-About how long does it take for P. elephas to mature?

Methods:

-The description of how eggs were weighed should occur before egg weight is mentioned (in section 2.3.)

-The literature that supports the Von Bertalanffy growth curve and aging lobster should be briefly described in the text so that readers can understand how the curve was formed. Age and growth of lobster are often uncertain. Previous literature suggests that Von Bertalanffy growth curves may not be appropriate for lobster since lobster do not grow continuously and growth varies among individuals (Chen et al. 2005).

-How are P. elephas aged?

-Briefly describe the Chen test.

-What is ELISA?

-Please elaborate on how size compositions were reported standardized to CPUE. This is not clear.

-What p-value is the threshold for significance in this study?

Results:

-The relationships between reproductive indicators and time should also be evaluated. Maturity and fecundity parameters seem to be evaluated over the entire time range, when they may have changed overtime in reality.

-Did fecundity-age relationships differ between specimens collected inside and outside each FPA?

-Did reproductive potential and fecundity differ between FPA area?

-How is the low values of RRP for small and large females related to poor representation? Wouldn’t poor representation lead to uncertain values, not necessarily low ones?

-Why is the catch of lobsters so much less in 2002-2003 for Su Pallosu?

-There seem to be more lobsters outside of the FPAs than instead the FPA (Figs. 9 and 10). Why would that be the case?

-Why is the catch of lobsters so much less in 2010 for Buggerru?

Discussion:

-The first paragraph of the discussion seems out of place. It seems more like introductory material.

-The discussion should have a section on how these methods could be applied to other fisheries.

-The discussion should also reflect more on the VTG analysis, since that was the novel method used in this study.

Minor comments:

Line 39- How many years were in the study?

Lines 41-42- These results do not suggest that an increase in FPAs can increase the inter-reserve connectivity.

Line 46- What is ‘halieutic’

Lines 47-50- Citation for this statement?

Line 59- What is the ‘maternal effect’?

Line 83- Citation for this statement?

Line 138- What is the ‘OUT zone’?

Lines 300-301- This sentence is not necessary since Table 2 has a table caption.

Figure 3- It looks like there are data from outside the FPA with CLs less than 80 mm, although the histogram in Figure 3c does not suggest there are.

Lines 322-324- A figure would help to get this point across.

Lines 354-356- This sentence is not clear. Doesn’t proportion and percentage mean the same thing in this case?

Line 385- What was the p-value?

Lines 385-386- This sentence is probably not necessary because Table 4 has a table caption.

Figures 9 and 10- I’m not sure what the benefit of these plots are to the paper. They seem more like supplementary material.

Lines 476-487- This sentence is vague.

Lines 487-491- Does the current study come to the same conclusions? If not, why?

Lines 491-493- Please elaborate how this study could separate the impact of restocking from the impact of the FPA.

Lines 497-502- Why would this be the case? I’m not following this logic. Please elaborate.

Lines 518-520- How can size at maturity be age-specific?

Lines 532-534- It’s not clear how the results in this study would lead to the conclusion that a FPA lasting more than 10 years could have a strong effect. I suggest removing this sentence.

Lines 585-586- Why would the recruitment of locally spawned larvae seem unlikely?

Reviewer 3 Report

General

11)      Why did the study focus only on females and not even acknowledge the importance of males? Past works have shown that in larger spiny lobsters there is significant sperm limitation and big males are critical for enabling more reproduction in populations. For examples and works within see:

https://www.tandfonline.com/doi/full/10.1080/17451000.2012.727429

https://www.jstor.org/stable/4601635

https://conbio.onlinelibrary.wiley.com/doi/10.1111/cobi.13535

https://academic.oup.com/icesjms/article/72/suppl_1/i115/618433

22)      The spatial arrangement of the study is not adequately portrayed. The reader is completely unsure as to where the nets were placed for both inside/outside and how that varied from year to year. Alternative explanations for variable catch should be at the least mentioned – do you have data for how temperature, ect related to the catch?

33) The hypotheses, and statistics used to address them, should follow a logical approach. I had a very hard time disentangling what the specific pairs of hypotheses and data were. It seems like many of the statements in the discussion and results are qualitative (i.e., the absolute # is > or <) versus statistically supported.

34)      The formatting and language both need substantial work to clarify. Figure fonts need to be increased so they can be read and to conform w journal req. Grammar needs checking. My review was hampered in part by unclear sentence structure, and I prioritized reviewing what I could understand as opposed to spending time deciphering meaning. Happy to review again once the language has been tightened up. The discussion is particularly hindered and challenging to review. I suggest creating an outline and perhaps including subheadings to rewrite the discussion in a focused manner that centers on the results of this study.

Specific

L48 – why only 2 decades?

L52 – caps

L52 – Why not just use the established acronym MPA throughout? Can specify the definition as it relates to your study. May help readers better relate to study vs FPA (which I have never encountered as an acronym for the same thing)

L56 to 60 – perhaps use reviews that synthesize each of these benefits.  Then go crustacean specific

L77 – great paragraph in terms of coverage. Language needs tightening

L92 – delete anyway, informal

L94 to 99 – Very confusing sentence. Rework it

L109 – previously it seemed like you were arguing that VTG is an established approach

Fig 1 – are the triangles the actual bounds of the MPAs? Bathymetry / Habitat available? Can we see where the fishing was? The outside and inside fishing would both be of extreme interest. Much could be done to give readers a better grasp of the study’s spatial coverage and significance

L123 – Cool approach. Was this supported by the fishers as well? Favorably received?

L140 – confusing

L140 – 0.1mm is perhaps too precise? How confident are the authors that this was the precision they could obtain? Even a slightly tilted caliper would throw off a 0.1 mm measurement. I find it unlikely that this is the appropriate level of sig figs

L151 | L172 – weird structure with many single sentence paragraphs

L207 – confusing

L220 – sample size is unclear form wording. Is this 60 total?

L259 – seems reasonable

L260 – why just females? Past works have shown that in larger Caribbean spiny lobsters there is significant sperm limitation and big males are critical for more productive populations.

Figure 2: is this a significant relationship? Equation?

Figure 3: mostly gorgeous figure. Needs font standardization. Cannot tell what the insets are. Doesn’t look like a sig change bw inside/outside which is good and interesting. The data in a/b should match that of c exactly, but it does not (i.e., there are 5 instances of fecundity > 100000 in c/d; but only 3 in a/b). Unsure why this is the case, but this is an urgent need to address for rigor.

Figure 4 – could combine into a single panel for space; note the %LD50 on the figure itself for clarity

Figure 5 – why are there so few legally fishable females in these areas?

Figure 7 – previously you defined CPUE as per net; why is it now per m? In either case, where is the effort compared and defined? More detail in fig1 would help tremendously

Fig 8 – b) based on the qualitative descriptions, it seems like the end result was not different from the start, yet it is also described differently. Slightly confused.

Figure 9 – what happened in 2002-2003? There is clearly something happening in terms of the spatial effort varying, or timing or gear or something. Dramatically different results via the ever important eyeball test. Id like to see this mentioned in the Discussion.

Figure 9/10 – unsure if these data are useful as presented; may be best placed into the supplementary materials

L403/4 – did the change in abundance ever get statistically quantified? If not, then much of the discussion needs to be refocused. Authors stat sig, but do not provide a test statistic nor p value

409 – looks to be a decreasing trend for the last 4 years of data

L429 – this paragraph is mostly introduction material. It has no mention of the work that was done. The discussion is typically where results are placed into context.

L449 – again, this belongs in the introduction as currently written.

L458 – this is a more relevant start to the discussion section, although it should be rewritten for clarity and concise

L466 – the phrasing here is off

L470 – structure

L475 – what are the growth rates, the maximum size, and other key life history information that may concurrently alter IEP?

L480 – structure

L501- both MPAs seem the same size in Fig 1.

L525 – structure

L584 – structure
